# Structural and Functional Impairments of Reconstituted High-Density Lipoprotein by Incorporation of Recombinant β-Amyloid42

**DOI:** 10.3390/molecules26144317

**Published:** 2021-07-16

**Authors:** Kyung-Hyun Cho

**Affiliations:** 1Korea Research Institute of Lipoproteins, Medical Innovation Complex, Daegu 41061, Korea; chok@yu.ac.kr; Tel.: +82-53-964-1990; Fax: +82-53-965-1992; 2LipoLab, School of Medical Biotechnology, Yeungnam University, Gyeongsan 38541, Korea

**Keywords:** apoA-I, beta-amyloid, high-density lipoproteins, Alzheimer’s disease, dementia

## Abstract

Beta (β)-amyloid (Aβ) is a causative protein of Alzheimer’s disease (AD). In the pathogenesis of AD, the apolipoprotein (apo) A-I and high-density lipoprotein (HDL) metabolism is essential for the clearance of Aβ. In this study, recombinant Aβ42 was expressed and purified via the pET-30a expression vector and *E.coli* production system to elucidate the physiological effects of Aβ on HDL metabolism. The recombinant human Aβ protein (51 aa) was purified to at least 95% purity and characterized in either the lipid-free and lipid-bound states with apoA-I. Aβ was incorporated into the reconstituted HDL (rHDL) (molar ratio 95:5:1, 1-palmitoyl-2-oleoyl-sn-glycero-3-phosphocholine (POPC):cholesterol:apoA-I) with various apoA-I:Aβ ratios from 1:0 to 1:0.5, 1:1 and 1:2. With an increasing molar ratio of Aβ, the α-helicity of apoA-I was decreased from 62% to 36% with a red shift of the Trp wavelength maximum fluorescence from 337 to 340 nm in apoA-I. The glycation reaction of apoA-I was accelerated further by the addition of Aβ. The treatment of fructose and Aβ caused more multimerization of apoA-I in the lipid-free state and in HDL. The phospholipid-binding ability of apoA-I was impaired severely by the addition of Aβ in a dose-dependent manner. The phagocytosis of LDL into macrophages was accelerated more by the presence of Aβ with the production of more oxidized species. Aβ severely impaired tissue regeneration, and a microinjection of Aβ enhanced embryotoxicity. In conclusion, the beneficial functions of apoA-I and HDL were severely impaired by the addition of Aβ via its detrimental effect on secondary structure. The impairment of HDL functionality occurred more synergistically by means of the co-addition of fructose and Aβ.

## 1. Introduction

Mild cognitive impairment (MCI), Alzheimer’s disease, and dementia are related to low serum HDL-C levels, particularly in middle age [1]. Conventionally, low serum HDL-C is an independent risk factor of metabolic syndrome, cardiovascular disease (CVD), and stroke [2]. Because almost 80% of patients with Alzheimer’s disease (AD) also have cardiovascular disease [3], the same common points between the two diseases are low serum HDL-C and dysfunctional HDL. 

The functionality of HDL is very important for suppressing the incidence of diabetes, cardiovascular disease, and stroke. More recently, coronary heart disease has been acknowledged as a potent risk factor for dementia [3,4]. Many cardiovascular risk factors, such as hypertension, diabetes mellitus, and dyslipidemia, are important risk factors of Alzheimer’s disease and vascular dementia [5]. Serum HDL can interact with β-amyloid (Aβ) in the brain [6]. Human apoA-I can bind Aβ and prevent Aβ-induced neurotoxicity [7]. 

Dysfunctional HDL and low serum apoA-I levels are a hallmark of diabetes, inflammation, and CVD [8]. A higher serum level of apoA-I and enhanced HDL functionality are critical for suppressing the incidence of CVD [9,10]. On the other hand, diabetes mellitus and AD are linked with the pathogenesis of advanced glycated end (AGE) products [11]. Glycated apolipoproteins are associated with a higher incidence of diabetes and Alzheimer’s disease [12], even though the mechanism has not been fully elucidated. 

The glycation of HDL also impaired its anti-inflammatory activity in innate immunity against viral infection through the loss of paraoxonase activity [13,14]. Native HDL showed potent antiviral ability against SARS-CoV-2, while glycated HDL lost its antiviral activity [13]. Many studies have shown that HDL functionality is strongly dependent on the composition of apolipoproteins and its glycation extent [15]. HDL functionality, including cholesterol efflux and anti-glycation activity, can be enhanced by aerobic exercise [16] and the consumption of functional foods [17]. 

HDL metabolism in the brain is basically different from that of blood and other organs. Brain cells can synthesize apo-E to form apo-E-enriched HDL (apo-E-HDL) but not apoA-I [18]. Serum apoA-I can cross the blood–brain barrier via a putative transporter to form apoA-I-HDL [19] and inhibit amyloid β (Aβ) aggregation [20]. On the other hand, no study has investigated the in vitro interactions of Aβ42 and apoA-I in the lipid-free and lipid-bound state and its physiological effect. 

In the current study, a reconstituted HDL containing apoA-I and Aβ42 was synthesized, and the functional and structural correlations of the rHDL with increasing Aβ content were characterized. The change in functionality was assessed by treating Aβ in serum HDL_3_ with fructose. The physiological functionality of apoA-I and the toxicity of the Aβ were evaluated using human macrophage cells, zebrafish embryos, and a wound-healing model of zebrafish. 

## 2. Results

### 2.1. Purification and Characterization of Aβ42

The recombinant Aβ42 (5.7 kDa) was expressed and purified to at least 95% purity from SDS-PAGE and densitometric scan analysis (Figure 1A). The amino acid sequence was NH_2_-MDAEFRHDSGYEVHHQKLVFFAEDVGSNKGAIIGLMVGGVVIA-L-E-HHHHHH-COOH. Protein prediction analysis showed that 48.8% was the beta-strand and 51.2% was the loop structure with a calculated isoelectric point of 5.59 at physiological pH. The Aβ42 was identified by immunodetection (Figure 1B) and amino-terminal protein sequencing. The extinction coefficient at 280 nm (ε_280_) was measured as 6350 M^−1^ cm^−1^ based on UV spectroscopy using Beckman DU800 spectrophotometer (Palo Alto, CA, USA) and a Suprasil quartz cuvette (1-cm path length). 

In the non-denaturing state, lipid-free Aβ showed an aggregated band pattern on the top of the loading position, as indicated by the red arrowhead (Figure 2A). In contrast, apoA-I showed a typical broadband pattern from 65 to 73 Å. In the lipid-bound state, the Aβ was condensed, and the aggregated band pattern was observed on the top of the loading position, whereas apoA-I in a lipid-bound state showed a major single band at 97 Å, as a black arrowhead. With an increasing amount of Aβ in the rHDL, the particle size gradually decreased up to 93 Å with a weaker smear intensity. This result indicates that Aβ interferes with the normal binding of apoA-I with phospholipid to form rHDL, although the solubility of Aβ was enhanced by the rHDL formation. Denaturing electrophoresis (18% SDS-PAGE) also showed that a more aggregated band appeared with increasing Aβ content in the rHDL (Figure 2B). Aβ in the lipid-bound state showed an aggregated band with lipid species and multimerization, as indicated in the red arrowhead. These results suggest that the addition of Aβ to the rHDL containing apoA-I severely impaired the stabilization of apoA-I in the rHDL

### 2.2. Structural Characteristics of rHDL Containing Aβ

The addition of Aβ severely interrupted the DMPC clearance activity of apoA-I. The phospholipid-binding ability was impaired severely by the addition of Aβ from the smallest amount. Native apoA-I alone showed a half-time of removal (T_1/2_) of 7 ± 1 min, as shown in Figure 3, but Aβ showed no DMPC binding ability and interrupted the binding ability of apoA-I. These results suggest that the addition of Aβ to the apoA-I-rHDL impaired the stabilization of apoA-I in rHDL. 

In the native state, apoA-I-rHDL showed 62.3% α-helicity from the CD analysis and the wavelength maximum fluorescence (WMF) at 337 nm, which are typical spectroscopic characteristics of apoA-I in the lipid-bound state. On the other hand, increasing the amount of Aβ in rHDL caused a decrease in α-helicity of 48.0%, 44.7%, and 36.2% for apoA-I:Aβ ratios of 1:0.5, 1:1, and 1:2, respectively (Table 1). WMF was also increased by 2–3 nm by the addition of Aβ (up to 340 nm, Table 1), indicating that Trp in apoA-I was more exposed to the aqueous phase by increasing the Aβ content. These results suggest that the incorporation of Aβ in rHDL resulted in the destabilization of the α-helical structure in apoA-I with a larger red shift of Trp, which contributed to the destabilization of apoA-I in rHDL.

### 2.3. Glycation of apoA-I and HDL_3_ Were Accelerated by Aβ

Treatment with fructose caused the severe glycation of apoA-I. The extent of glycation increased depending on the amount of Aβ added over 144 h (Figure 4). Treatment of fructose or co-treatment of fructose and Aβ with apoA-I caused up to 13-fold or 26-fold more severe glycation, respectively, than apoA-I alone. Interestingly, a mixture of apoA-I and Aβ also showed 5.4-fold higher fluorescence intensity than apoA-I alone, indicating that Aβ could facilitate the glycation without fructose.

The glycation resulted in the multimerization of apoA-I (Figure 5) during the 144 h incubation. Interestingly, the addition of fructose caused the disappearance of the apoA-I band (lane 2, Figure 5A,B) with more aggregation in the absence of an Aβ treatment. Co-treatment of Aβ and fructose caused a larger decrease in apoA-I in a dose-dependent manner up to 72 μM (lanes 4, 5, and 6 of Figure 5). The apoA-I band size was decreased up to 0.61 upon addition of Aβ (final 72 μM) compared with apoA-I alone with a band size of 1.00. These results suggest that the incorporation of Aβ can degrade apoA-I synergistically in the presence of fructose.

The addition of fructose to HDL_3_ also caused an increase in yellow fluorescence (Figure 6A) and multimerization of apoA-I (lane 2 of Figure 6B). The non-enzymatic glycation of HDL_3_ was also accelerated by increasing the Aβ level with the concomitant disappearance of the apoA-I band and multimerization of apoA-I (lane 4, 5, 6 in Figure 6B) over 144 h.

### 2.4. Aβ Caused More Rapid Isothermal Denaturation of rHDL

In the presence of urea, rHDL was resistant to the denaturation stress without an increase in WMF until 4 M urea addition, as shown in Table 2 and Appendix A. On the other hand, the addition of Aβ resulted in a 3–4 nm increase in WMF in the presence of urea from 0 to 4 M; rHDL (apoA-I:Aβ, 1:0) showed 338.5 and 340.7 nm at 0 and 4 M urea, respectively, while rHDL (apoA-I:Aβ, 1:2) showed 341.5 and 344.0 nm at 0 and 4 M urea, respectively. All rHDLs showed similar WMF values around 346–347 nm regardless of Aβ content, in the presence of 5 M urea, which represents the late stages of the denaturation process. This result indicates that the destabilization of apoA-I by means of the addition of Aβ occurred distinctly at low concentrations of urea of around 1–2 M, representing the early stages of the denaturation process.

### 2.5. More LDL Phagocytosis into Macrophage by Aβ

In the absence of oxLDL, apoA-I-rHDL treated cells showed a higher cell number (3.1 × 10^5^ cells, photo b) than the control (2.6 × 10^5^ cells, photo a) from morphological observation, as shown in Figure 7A. On the other hand, an increase in the Aβ content in the rHDL caused a gradual decrease in cell number, leading to a 55% decrease in the number of cells (1.4 × 10^5^ cells, photo e) due to rHDL treatment (apoA-I:Aβ, 1:2). This result shows that the cytoprotective properties of rHDL (apoA-I:Aβ, 1:0) can be impaired by the increase in Aβ content in the rHDL up to apoA-I:Aβ 1:2. Taken together, the incorporation of Aβ into HDL resulted in the production of dysfunctional HDL which is more susceptible to denaturation and cytotoxicity. Similarly, another group also showed that Aβ treatment induced the proinflammatory activation of THP-1 cells and the up-regulation of interleukin (IL)-6 and IL-1β with cytotoxicity [21].

The phagocytosis of oxLDL (photo f in Figure 7A) into macrophages caused the production of more oxidized species (Figure 7B) and cell death (2.3 × 10^5^ cells) in the absence of rHDL. Co-treatment of the apoA-I-rHDL caused less cell death (2.4 × 10^5^ cells, photo g) than the oxLDL treatment without producing more oxidized species. On the other hand, an increase in the Aβ ratio in the rHDL caused a gradual decrease in cell number, leading to a 46% decrease in the number of cells (1.3 × 10^5^ cells, photo e) by means of treatment with rHDL containing Aβ (apoA-I:Aβ, 1:2), with the production of more oxidized species up to 13 nM of MDA. 

### 2.6. Toxicity on Embryo and Tissue Regeneration of Aβ

As shown in Figure 8, an injection of Aβ attenuated the developmental speed of embryos and a higher death rate in a dose-dependent manner. The highest dose of Aβ (3.6 μM) resulted in the lowest embryo survivability (~59%). Aβ (final 1.8 μM) injection resulted in 40% death of the embryos, while the same concentration of apoA-I injection resulted in 19% embryo death 24 h post-injection. 

As shown in Figure 9, a comparison of the tissue regeneration effect showed that an apoA-I (final 36 μM) injection caused 11% faster regeneration activity than the Tris-buffer control. In comparison, an Aβ (final 36 μM) injection caused 40% lower regeneration activity than the control. These results suggest that Aβ is more toxic to embryos and impairs tissue regeneration compared to plasma apoA-I.

## 3. Discussion

HDL has emerging roles in neurodegenerative disorders because cholesterol transport in the brain is strongly dependent on the HDL metabolism. Higher plasma levels of HDL and apoA-I are correlated directly with a lower risk of developing AD and dementia [22]. Although the accumulation of Aβ is a major culprit of AD, reduced serum apoA-I has been detected in AD [23]. The brain tissue has the highest cholesterol content (~15 mg/g tissue), while the adrenal gland, lung, and average tissue show 9, 6, and 2–3 mg/g tissue, respectively, in a mouse study [24]. The half-life of cholesterol in the brain is much longer (six months to five years) than that of the blood cholesterol (a few days) [25]. Brain cells can synthesize their own cholesterol; however, cholesterol in the blood, mainly synthesized in the liver, cannot be delivered into the brain tissue. Because LDL cannot cross the blood–brain barrier (BBB), cholesterol in the blood and brain is separated by the BBB. 

Instead of LDL, HDL-like particles containing apo-E can be synthesized in neuronal and glial cells of the brain [26]. The major carrier of cholesterol in the brain is apo-E-HDL because the brain tissue can synthesize apo-E but cannot synthesize apo-B and LDL [27]. On the other hand, apoA-I and discoidal HDL in blood can be transported into brain tissue via the clathrin-independent and cholesterol-mediated endocytosis pathway [28]. However, apo-E, apo-B, and LDL cannot cross BBB, because the brain is isolated and protected by BBB from pathogens in the blood.

The beneficial role of apoA-I and apoA-I-HDL in the brain to prevent amyloid aggregation is still unclear, even though a clinical report showed that a lower serum HDL-C is associated with a higher risk of AD [1,29]. On the other hand, there is limited information on the binding process of Aβ and HDL at the protein level on the neuron system. HDL has a protective effect against the incidence of Alzheimer’s disease by preventing amyloid aggregation because HDL and apoA-I can bind with the amyloid protein directly [29] to remove amyloid plaque. ApoA-I also binds Aβ to prevent the induction of neurotoxicity [20]. Although the precise interaction mechanism of apoA-I and Aβ has not been elucidated, an intravenous injection of rHDL containing apoA-I reduced the soluble brain Aβ in a symptomatic mice model [29]. 

The current study showed the physiological role of purified Aβ (Figure 1) upon binding with apoA-I and HDL. The structural and functional features of apoA-I can be impaired severely by Aβ upon binding in rHDL; a smaller particle size of rHDL was produced with more binding of Aβ in rHDL (Figure 2). The addition of Aβ caused the impairment of the phospholipid-binding ability of apoA-I (Figure 3) and decrease in α-helix content in apoA-I in the rHDL state (Table 1). With increasing Aβ content, the glycation of HDL was further accelerated (Figure 4) with the degradation and multimerization of apoA-I (Figure 5 and Figure 6). This result strongly suggests that there is a strong synergistic effect of Aβ to initiate glycation reaction. It has been shown that Lys-16 and Arg-5 in the primary sequence of Aβ are critical to initiate glycation in the presence of oxidative stress, as previously reported [30]. With increasing Aβ content in rHDL, Trp was more exposed to aquatic phase as determined by isothermal denaturation (Table 2). Taken together, the presence of Aβ caused a detrimental effect on the HDL structure via interference in the conformational stability by apoA-I. Atherogenic phagocytosis of oxLDL into macrophages was accelerated more by the addition of Aβ Figure 7). An injection of Aβ resulted in toxicity to embryos (Figure 8) and impaired tissue regeneration activities (Figure 9) in a zebrafish model. 

In addition to diabetes, the AGE levels are higher in patients with AD [31]. The glycation of apoA-I is a pathological process of diabetes, CVD, and neurodegenerative disease [32]. These results suggest that Aβ can facilitate more glycation of apoA-I to exacerbate AD and CVD. In addition to the high level of glycation, a higher serum level of trans fat was closely associated with the risk of dementia, according to Hisayama’s study [33]. apoA-I-rHDL containing elaidic acid showed impaired HDL functionality via the displacement of apoA-I from the rHDL [32]. Several atherogenic changes in HDL by carbohydrate and trans fat HDL show that good-quality HDL reduces the risk of AD via the natural clearance of Aβ [34].

## 4. Materials and Methods

### 4.1. Materials

Cloned gene of amyloid beta (pCNS-Aβ-cDNA), Cat # KU000776) was obtained from the Human Gene Bank of Korea (Daejeon, Korea). The pET30a(+) expression vector and *E.coli* BL21 (DE3) were purchased from Novagen (Madison, WI, USA). The restriction enzymes were acquired from New England BioLabs (Beverly, MA, USA). Palmitoyloleoyl phosphatidylcholine (POPC, #850457) was supplied by Avanti Polar Lipids (Alabaster, AL, USA). Sodium cholate (#C1254) was procured from Sigma (St. Louis, MO, USA).

### 4.2. Expression and Purification of Aβ

The human Aβ gene was cloned using a polymerase chain reaction (PCR) with the forward primer 5′-ATG GTACATATGGATGCAGAATTCCGACATG-3′ and reverse primer 5′-GTGTTGTCATAGCGCTCGAGATGGTA-3′ to generate *Nde I* and *Xho I* sites to construct the expression vector.

The subcloned cDNA was inserted into the pET30a expression vector to be verified by DNA sequencing using a Sequentator (ABI7500, ABI, Foster City, CA, USA). The expressed polypeptide has 51 amino acids containing a 1-start codon and 8-His tag. The mature form was 42 amino acids, and the His-tag (8 amino acids, L-E-HHHHHH) in the C-terminal and Met in the N-terminal were used as the start codon. The His-tagged Aβ gene was expressed and purified by Ni^2+^-nitrilotriacetic acid chromatography column chromatography (Peptron, Cat#1103-3, Daejeon, Korea). The fractions containing Aβ were pooled and dialyzed against a buffer containing 10 mM Tris-HCl (pH 8.0) and 10% glycerol. The protein concentration was determined by a Bradford assay using bovine serum albumin (BSA) as a standard. The protein purity was initially monitored by SDS-PAGE and Coomassie blue staining. 

### 4.3. Protein Sequencing

Protein samples for sequencing were electrotransferred onto a PVDF membrane (Immobilon-P) using the protocol outlined by Matsudaira [35]. The NH_2_-terminal amino acid sequence of the excised band was determined using an Applied Biosystems model 491A sequencer (Foster City, CA, USA) located in the Korea Basic Research Institute (Daejeon, Korea).

### 4.4. Characterization of Secondary Structure 

The average α-helix contents of the proteins in the lipid-free and lipid-bound states were measured by circular dichroism (CD) spectroscopy (J-700, Jasco, Tokyo, Japan). The spectra were obtained from 250 to 190 nm at 25 °C in a 0.1 cm path-length quartz cuvette at a bandwidth of 1.0 nm, a speed of 50 nm/min, and a response time of 4 s. Samples of lipid-free and lipid-bound proteins were diluted to 0.07 mg/mL to avoid self-association, whereas lipid-bound proteins were diluted to 0.1 mg/mL. Four scans were accumulated and averaged. The α-helical content was calculated from the molar ellipticity at 222 nm [36]. 

### 4.5. Characterization of Trp Fluorescence during Isothermal Denaturation

The wavelengths of maximum fluorescence (WMF) of the tryptophan (Trp) residues in apoA-I were determined from the uncorrected spectra using an LS55 spectrofluorometer (Perkin-Elmer, Norwalk, CT, USA), as described previously [37], using WinLab software package 4.00 (Perkin-Elmer) and a 1 cm path-length Suprasil quartz cuvette (Fisher Scientific, Pittsburgh, PA, USA). The samples were excited at 295 nm to avoid tyrosine fluorescence, and the emission spectra were scanned from 305 to 400 nm at room temperature [38]. For isothermal denaturation, the effects of urea addition on the secondary structures of Aβ and apoA-I in a lipid-bound state were monitored by measuring the α-helicity and tryptophan movement using CD and fluorospectroscopy, as reported elsewhere [36,37,38].

### 4.6. Purification of Lipoproteins

LDL (1.019 < d < 1.063), HDL_2_ (1.063 < d < 1.125), and HDL_3_ (1.125 < d < 1.225) were isolated from the sera of young and healthy human males (mean age, 22 ± 2 years, *n* = 20), who voluntarily donated blood after fasting overnight via sequential ultracentrifugation; the density was adjusted appropriately by adding NaCl (Sigma # S9888)and NaBr (Sigma # 310506), as detailed elsewhere [39], and procedures were carried out in accordance with standard protocols [40]. The samples were centrifuged for 24 h at 10 °C at 100,000× *g* using a Himac CP-100NXα (Hitachi, Tokyo, Japan) at the LipoLab of Yeungnam University. After centrifugation, each lipoprotein sample was dialyzed extensively against Tris-buffered saline (TBS; 10 mM Tris-HCl, 140 mM NaCl, and 5 mM EDTA (pH 8.0)) for 24 h to remove NaBr.

### 4.7. Purification of Human apoA-I

ApoA-I was purified from HDL by ultracentrifugation, column chromatography, and organic solvent extraction according to the method described by Brewer et al. [41]. At least 95% protein purity was confirmed by SDS-PAGE.

### 4.8. Oxidation of LDL

Oxidized LDL (oxLDL) was produced by incubating the LDL fraction with CuSO_4_ (Sigma # 451657) final concentration, 10 μM for 4 h at 37 °C. OxLDL was then filtered (0.22 μm filter) and analyzed using a thiobarbituric acid reactive substances (TBARS) assay to determine the extent of oxidation with a malondialdehyde (MDA, Sigma # 63287) standard, as described previously [42].

### 4.9. Synthesis of Reconstituted HDL

Reconstituted HDL (rHDL) was prepared using the sodium cholate dialysis method [43] at an initial molar ratio of 95:5:1:0, 95:5:1:0.5, 95:5:1:1, and 95:5:1:2 for POPC:cholesterol:apoA-I:Aβ, respectively. The size and hydrodynamic diameter of the rHDL particles were determined by 8–25% native polyacrylamide gradient gel electrophoresis (PAGGE, Pharmacia Phast system) by a comparison with the standard globular proteins (GE Healthcare, Uppsala, Sweden). After dialysis, all rHDL showed a similar range of residual endotoxin levels, less than 3.1 to 3.3 EU/mL, based on endotoxin quantification using a commercially available test kit (BioWhittaker, Walkersville, MD, USA).

### 4.10. Phospholipid Binding Assay

The interactions of the apoA-I and Aβ with 1,2-dimyristoyl-sn-glycero-3-phosphocholine (DMPC, Avanti Polar Lipids, Cat #850345) were monitored using a slight modification of the method described by Pownall et al. [44]. The DMPC to protein mass ratio was 2:1 (*w*/*w*), in a total reaction volume of 0.76 mL. The measurements were initiated after adding DMPC and monitored at 325 nm every 2 min using an Agilent 8453 UV-visible spectrophotometer (Agilent Technologies, Waldbronn, Germany) equipped with a thermocontrolled cuvette holder adjusted to 24.5 °C.

### 4.11. Glycation of apoA-I with Aβ

The glycation sensitivity was compared by incubating the purified lipid-free apoA-I (final 1 mg/mL) with 250 mM D-fructose (Sigma # F2793) in 200 mM potassium phosphate/0.02% sodium azide buffer (pH 7.4), as reported elsewhere [45]. ApoA-I was incubated for up to 144 h in an atmosphere containing 5% CO_2_ at 37 °C. The extent of the advanced glycation reactions was determined by reading the fluorescence intensities at 370 (excitation) and 440 nm (emission), as described previously [46].

### 4.12. Western Blotting

Aβ protein (3 µg) of the lipid-free state was loaded and electrophoresed on 15% SDS-PAGE gels and detected by the Aβ antibody (ab62658, Abcam, London, UK) as the first antibody (diluted 1:2000) and goat anti-rabbit immunoglobulin G-horseradish peroxidase (HRP) (A120-101P, Bethyl Laboratories, Montgomery, AL, USA) as the secondary antibody (diluted 1:5000). The ApoA-I and Aβ concentration in the lipid-free states was determined using the Bradford assay modified by Markwell et al. [47] using bovine serum albumin as a standard.

### 4.13. LDL Phagocytosis Assay

THP-1 cells, a human monocyte cell line, were obtained from the American Type Culture Collection (ATCC, #TIB-202™; Manassas, VA, USA) and maintained in RPMI1640 medium (Hyclone, Logan, UT, USA) supplemented with 10% fetal bovine serum (FBS) until required for experimentation. The cells that had undergone no more than 20 passages were incubated in a medium containing phorbol 12-myristate 13-acetate (PMA; final 150 nM, Sigma #P8139) in 24-well plates for 48 h at 37 °C in a humidified incubator (5% CO_2_, 95% air) to induce differentiation into macrophages. The differentiated and adherent macrophages were incubated with fresh RPMI-1640 medium containing 1% FBS, oxidized LDL (oxLDL; 50 µg), and each protein (final concentration, 1.8, 3.6, 7.2 µM) for 48 h at 37 °C in a humidified incubator. After incubation, the cells were washed three times with PBS and fixed in 4% paraformaldehyde (Sigma # 818715) for 10 min. The fixed cells were then stained with an oil-red O staining (Sigma # O0625) solution (0.67%) and washed with distilled water. The THP-1 macrophage-derived foam cells were then observed and photographed using a Nikon Eclipse TE2000 microscope (Tokyo, Japan) at 600× magnification.

### 4.14. Zebrafish

Zebrafish and embryos were maintained using standard protocols. The maintenance of zebrafish and procedures using zebrafish were approved by the Committee of Animal Care and Use of Yeungnam University (Gyeongsan, Korea). The fish were maintained in a system cage at 28 °C during treatment under a 10:14 h light cycle with the consumption of normal tetrabit (TetrabitGmbh D49304, 47.5% crude protein, 6.5% crude fat, 2.0% crude fiber, 10.5% crude ash, containing vitamin A (29,770 IU/kg), vitamin D3 (1860 IU/kg), vitamin E (200 mg/kg), and vitamin C (137 mg/kg); Melle, Germany).

### 4.15. Microinjection of Zebrafish Embryos

Embryos at one-day post-fertilization (dpf) were injected individually by microinjection using a pneumatic picopump (PV820; World Precision Instruments, Sarasota, FL, USA) equipped with a magnetic manipulator (MM33; Kantec, Bensenville, IL, USA) with a pulled microcapillary pipette-using device (PC-10; Narishigen, Tokyo, Japan). To minimize bias, the injections were performed at the same position on the yolk. Up to 200 ng of Aβ or apoA-I in the lipid-free state were injected into flasks of embryos (final amount, 50 nL). After the injection, live embryos were observed under a stereomicroscope (Motic SMZ 168; Hong Kong) and photographed using a Motic cam2300 CCD camera.

### 4.16. Fin Regeneration

The wound-healing effect of apoA-I and Aβ was tested using adult zebrafish. The experimental zebrafish, approximately 12 weeks old, were anesthetized by submersion in 2-phenoxyethanol (Sigma P1126; St. Louis, MO, USA) in system water (1:1000 dilution). For the fin regeneration studies, the zebrafish were anesthetized, and the tail fins were cut with a scalpel close to the proximal branch point of the dermal rays within the fin. After amputation, 10 µL of each protein Aβ and apoA-I in the lipid-free state (final 34 µM of protein) was injected into the tail muscle near the urostyle (*n* = 7 for each group). After the injection, the fish consumed a normal diet and were observed in a 28 °C system incubator. Images of the regenerating fins from live zebrafish were taken at 24 h intervals, up to 144 h, under a stereomicroscope (Motic SMZ 168; Hong Kong) and photographed using a Motic cam2300 CCD camera, using Image Proplus software version 4.5.1.22 (Media Cybernetics, Bethesda, MD, USA).

### 4.17. Statistical Analysis

The data in this study were expressed as the mean ± SD from at least three independent experiments with duplicate samples. For the zebrafish study, multiple groups were compared using one-way analysis of variance (ANOVA) between the groups using Scheffe test. Statistical analysis was performed by SPSS software program (version 23.0; SPSS, Inc., Chicago, IL, USA). A *p* value < 0.05 was considered statistically significant. 

## 5. Conclusions

The incorporation of Aβ caused the aggregation of apoA-I and a change in dysfunctional HDL, with the loss of anti-atherogenic activity and toxicity to embryo development and tissue regeneration. These results might explain how the accumulation of Aβ can exacerbate the progress of AD and CVD via an interaction with HDL.

## Figures and Tables

**Figure 1 molecules-26-04317-f001:**
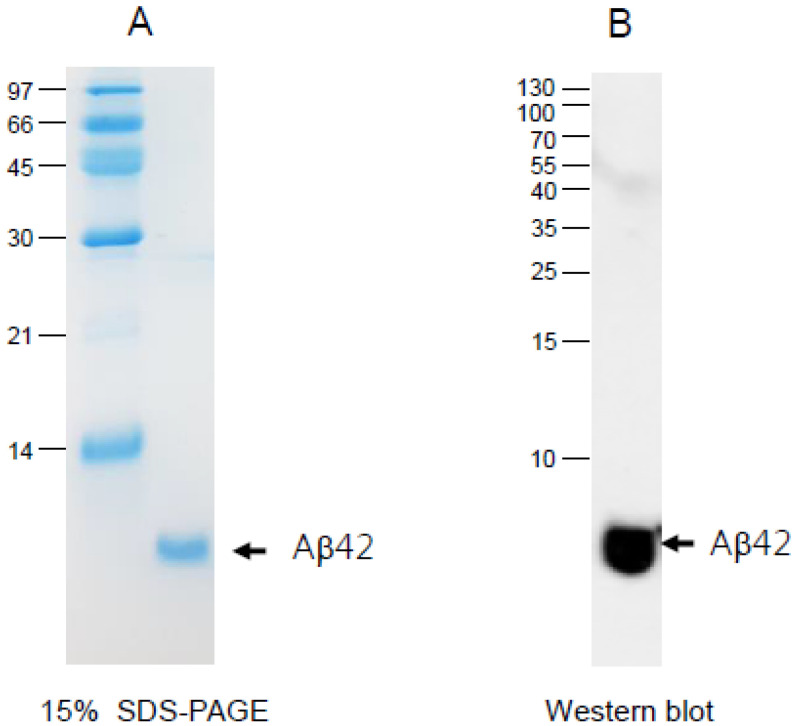
Electrophoretic patterns of purified Aβ42 (**A**) and identification by Western blot (**B**).

**Figure 2 molecules-26-04317-f002:**
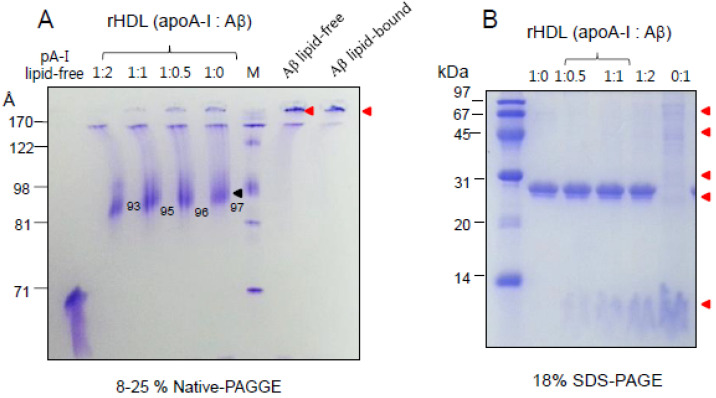
Electrophoretic patterns of apoA-I-rHDL containing Aβ with different molar ratios of apoA-I:Aβ (1:0, 1:0.5, 1:1, 1:2). (**A**) Each rHDL was migrated on an 8–25% native polyacrylamide gradient gel electrophoresis (PAGGE) without denaturation of proteins. Black arrowhead indicates major band of rHDL containing apoA-I. (**B**) Electrophoretic patterns of each rHDL with increasing amount of Aβ (18% SDS-PAGE). Red arrowheads indicate aggregated bands of Aβ.

**Figure 3 molecules-26-04317-f003:**
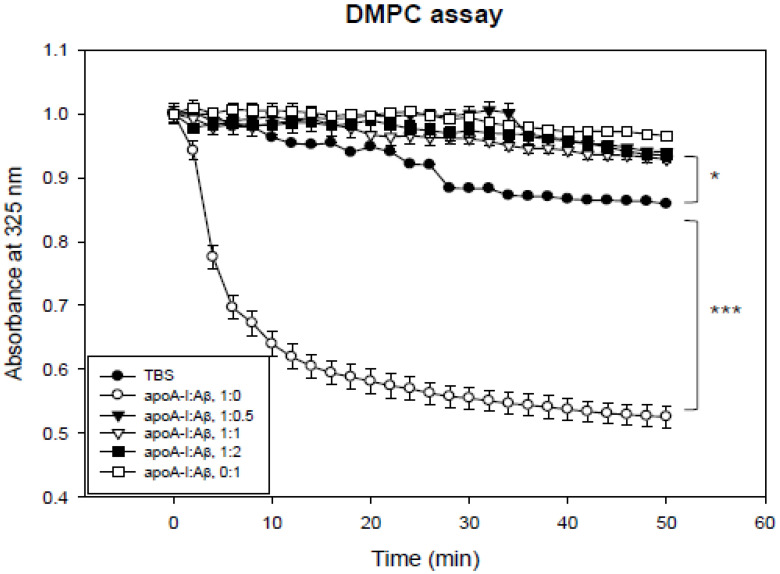
Kinetics of apoA-I and beta-amyloid (Aβ) with DMPC multilamellar liposomes. The absorbance at 325 nm was monitored at 24.5 °C at 2 min intervals. *, *p* < 0.05 between TBS (closed circle) and apoA-I:Aβ, 1:2; ***, *p* < 0.001 between TBS and apoA-I:Aβ, 1:0 (open circle).

**Figure 4 molecules-26-04317-f004:**
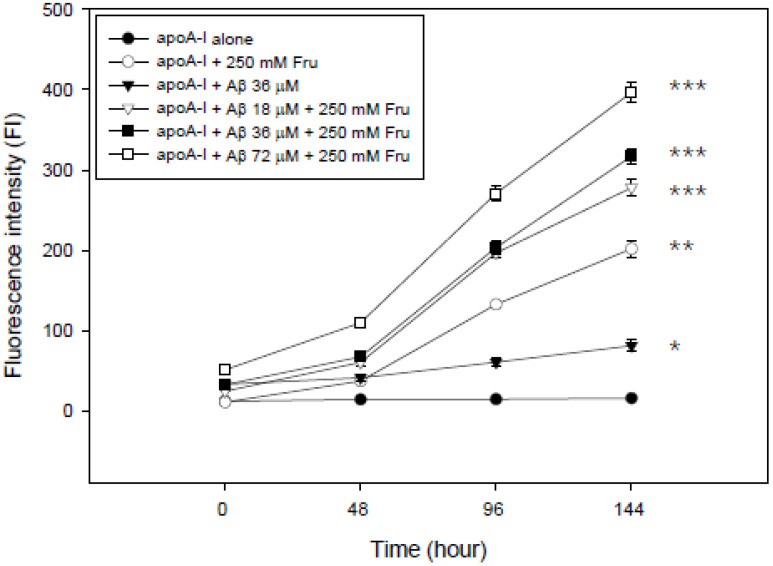
Synergistic increase in the extent of glycation by elevation of the Aβ content over 144 h in the presence of fructose (Fru, final 250 mM). *, *p* < 0.05 between apoA-I alone and apoA-I + Aβ 36 μM; **, *p* < 0.01 between apoA-I alone and apoA-I + 250 mM Fru; ***, *p* < 0.001 between apoA-I alone and apoA-I + Aβ + 250 mM Fru.

**Figure 5 molecules-26-04317-f005:**
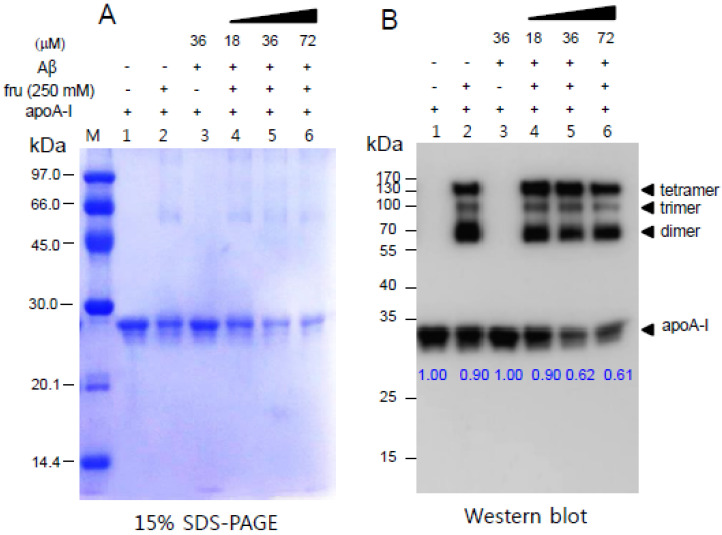
Degradation of apoA-I in the lipid-free state by a treatment with Aβ in the presence of fructose as visualized by Coomassie Brilliant Blue staining (**A**) and immunodetection with apoA-I antibody (**B**). Blue numbers below apoA-I band indicate a band size of apoA-I monomer in each lane from image analysis.

**Figure 6 molecules-26-04317-f006:**
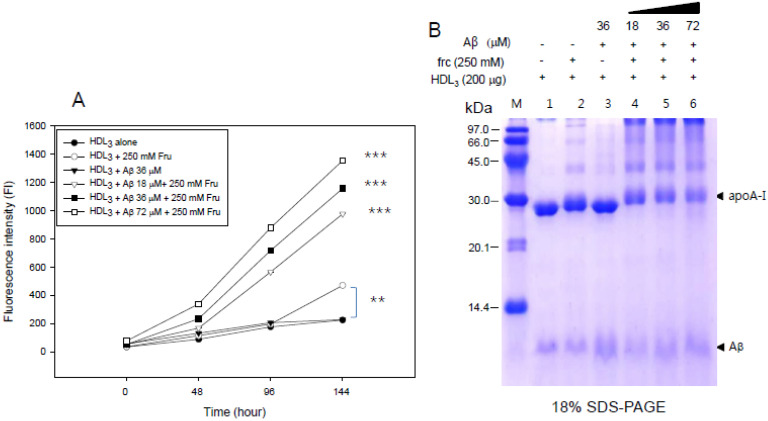
Synergistic increase in the extent of HDL glycation depends on the Aβ content in the presence of fructose. (**A**) A larger increase in fluorescence was detected in HDL with a larger increase in Aβ in relation to the concentration of fructose (**A**). **, *p* < 0.01 compared with HDL_3_ alone; ***, *p* < 0.001 compared with HDL_3_ alone. More multimerization and degradation of apoA-I were detected by Coomassie Brilliant Blue staining (**B**).

**Figure 7 molecules-26-04317-f007:**
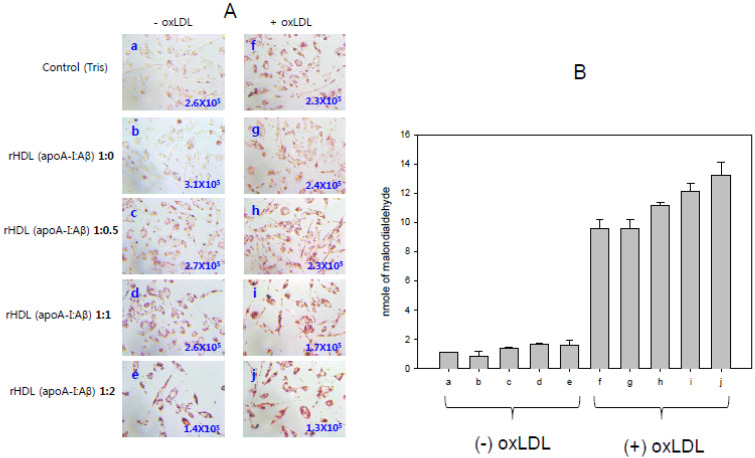
Phagocytosis of oxLDL into macrophages in the presence of rHDL containing apoA-I and Aβ. Lipid accumulation was visualized by Oil–red O staining (**A**). Quantification of oxidized species in the cell media by a TBARS assay (**B**).

**Figure 8 molecules-26-04317-f008:**
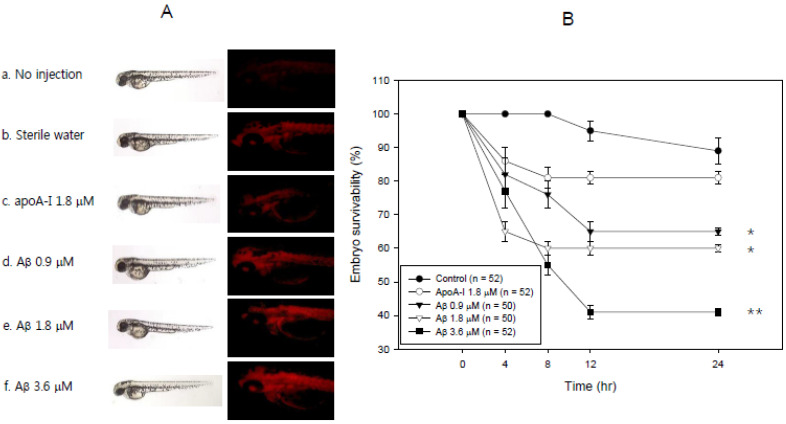
Comparison of the toxicity to embryos between apoA-I and Ab in zebrafish embryo. (**A**) Developmental status of embryo and visualization of ROS by DHE staining. (**B**) Survivability of embryos during the 24 h post-injection with apoA-I or Aβ. *. *p* < 0.05; **, *p* < 0.01 compared with apoA-I at 24 h incubation time.

**Figure 9 molecules-26-04317-f009:**
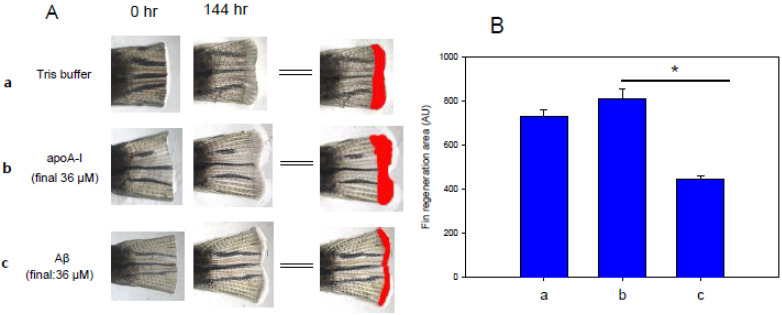
Comparison of tissue regeneration ability between apoA-I and Aβ. (**A**) Representative image of tail fin regeneration pattern at 144 h post-injection of apoA-I or Aβ (final 36 μM). (**B**) Image analysis of fin regeneration area at 144 h after protein injection. The data are shown as the mean ± standard deviation (SD) of three independent experiments (*n* = 7). *, *p* < 0.05 between apoA-I and Aβ.

**Table 1 molecules-26-04317-t001:** Spectroscopic characterization of rHDL containing apoA-I and Aβ in the lipid-bound and lipid-free state.

	Molar Composition (POPC:FC:apoA-I:Aβ)	WMF ^a^ (nm)	α-Helicity ^b^ (%)	Size (Å) ^c^
apoA-I-rHDL	95:5:1:0	337.5	62.3	97
rHDL (apoA-I:Aβ)	95:5:1:0.5	339.2	48.0	96
	95:5:1:1	339.4	44.7	95
	95:5:1:2	340.6	36.2	93
	95:5:0:1	ND	10.5	ND

^a^ Determined by circular dichroism spectroscopy. ^b^ Determined by fluorospectroscopy (Ex = 295 nm, Em = 310–400 nm). ^c^ Determined by SDS-PAGE and densitometric analysis. POPC, palmitoyloleoyl phosphatidylcholine; WMF, wavelength maximum fluorescence; rHDL, reconstituted high-density lipoproteins. ND, not detected.

**Table 2 molecules-26-04317-t002:** Wavelength maximum fluorescence (Ex = 295 nm, Em = 310–400) of rHDL containing apoA-I and Ab in the presence of urea for isotheral denaturation.

	0 M Urea	1 M Urea	2 M Urea	3 M Urea	4 M Urea	5 M Urea
Lipid-free apoA-I	339.7 ± 0.4	343.8 ± 1.0	347.8 ± 0.1	353.8 ± 0.4	355.3 ± 0.5	355.7 ± 1.0
rHDL- (apoA-I: Aβ, 1:0)	338.5 ± 0.3	340.3 ± 0.5	340.5 ± 0.7	338.7 ± 0.3	340.7 ± 0.5	347.2 ± 0.7
rHDL- (apoA-I:Aβ, 1:0.5)	339.5 ± 0.3	339.0 ± 0.6	341.5 ± 0.3	342.2 ± 0.6	344.4 ± 0.4	346.2 ± 0.1
rHDL- (apoA-I:Aβ, 1:1)	339.0 ± 0.1	340.0 ± 0.7	339.3 ± 0.2	342.0 ± 0.1	345.0 ± 0.4	346.2 ± 0.2
rHDL- (apoA-I:Aβ, 1:2)	341.5 ± 0.3	343.0 ± 0.4	344.0 ± 1.1	344.0 ± 0.1	344.0 ± 0.3	346.0 ± 0.4

rHDL, reconstituted high-density lipoproteins; apoA-I, apolipoprotein A-I; Aβ, amyloid beta.

## Data Availability

The data used to support the findings of this study are available from the corresponding author upon reasonable request.

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
