# Peer review of "Structural and Functional Impairments of Reconstituted High-Density Lipoprotein by Incorporation of Recombinant β-Amyloid42"

_molecules, 2021, doi:10.3390/molecules26144317_

Round 1
Reviewer 1 Report
This a nicely conceptualised and contextualised study, looking at a potentially important area. However, I have some concerns, which are listed below:
i) Is there a conflict between the reference [6], and the paragraph opening at line 56?
ii) A functional assay for the reconstituted HDL should be included - e.g. a classic 'efflux' assay, to show equivalence to 'normal' HDL
iii) Statistics - only a Student's t-test cited: ANOVA and post-tests should be needed for multiple comparisons
iv) Figure 2: 'Beta' is missing in the legend in two places
v) Figure 3: Error bars are shown, but no statistics have been performed - is this a single experiment? Further - the text (line 246/247) indicates dose dependency, which is not clearly delineated in this figure.
vi) Figure 5: Again, is this n=1? No statistics have been utilised to support the claims in the text. No obvious dose dependency when cf. 18 microM and 36 microM, despite claims in the text
vii) Figure 6: Again, n=1? Figure 7 n=1? The reproducibility of the findings needs to be supported by further experimentation.
viii) The authors cite 'growth' of THP-1 macrophages. Differentiation with phorbol ester halts 'growth' of this cell line. An assay for viability or toxicity needs to be employed to support the claims in the text.
ix) Figure 9: Again, the numbers of experiments and embryos examined needs to be detailed.
x) Figure 10: This is the only figure where experimental reproducibility seems to have been demonstrated, and the only one where statistical significance has been shown.
Author Response
Thank you very much for your valuable comments.
Please find attached doc. file for point-to-point response

Reviewer 2 Report
The manuscript by Prof. Cho explored an important relationship that exist, and often underestimated, between HDL (and ApoAI) and Amyloid beta. The results presented suggest that A-beta destabilizes ApoAI in rHDL and this effect could influence the various properties that the particle possesses in vivo. Although clear, the results can be improved. For this reason, there are major issues (procedure/results related) and minor issues (mainly related to language errors) that should be addressed.
Major issues
1) the paragraph from line 230 to 240 is somewhat confusing and I don’t know if the Angstrom unit is an error or not. In fact, from the SDS-PAGE you can just determine the apparent molecular weight of the molecule, whereas its size (hydrodynamic radius, so the dimension) can be estimated for instance by gel filtration chromatography. Please correct Angstrom to kDa if it is an error, otherwise explain better. The same is true for Table 1, which reports “Size (Angstrom)”.
2) Lines 235-236. It is somewhat strange that the molecular weight of the rHDL decreases with increasing ratios of ApoAI-Abeta. In particular considering that Abeta solubility seems increased in the presence of rHDL since the band above 170 kDa shows a decreased intensity with increasing concentrations of Abeta. How do you comment on that? Is there any residual protease activity that copurify with the protein?
3) Regarding the results of figure 2B, how did you conclude that Abeta destabilize ApoA-I in rHDL? From what I see, by increasing the ratio of Abeta, there is a smear increasing its intensity below 14 kDa, which should be multimerized Abeta (or perhaps aggregated forms). However, the intensity of the band around 30 kDa form ApoA1 seems not altered nor aggregated. You should do a western blot to confirm that the smear below 14 kDa is Abeta. Indeed, it is possible that the smear is due to some kind of association or co-localization of lipids in the gel. Nonetheless, I think that you should review what your results suggest.
4) From what I see from the results on figure 3, there is not really a concentration dependence of ApoAI activity impairment by Abeta. Indeed, as soon as Abeta is added, the activity of ApoAI is impaired. Perhaps the concentration of Abeta is too high to appreciate a subtle impairment on the activity of ApoAI.
5) the results on Figure 4 are quite strange. In the presence of A-beta, but without fructose, the fluorescence of the formed pterin increases. Why is so?
6) From the results of Figure 5, it seems that ApoAI monomer concentration decreases by increasing the time of incubation in the presence of both Fructose and Abeta (except the last lane, where I suppose there is some sort of edge effect in the transfer of proteins to the membrane). However, it is difficult to detect a real change in the intensities because: a) there is no histogram showing intensities change; b) the western blot seems a bit saturated. I suggest to obtain a figure with a lower saturation and to provide the histogram showing intensities. In addition, if the amount of ApoAI decreases, where did the protein go? If the samples are free of protease, it could be a precipitation of the protein. You should comment on this because this experiment shows that A-beta could decrease the stability of the ApoAI as well as increase its susceptibility to glycative stress.
7) From the result presented on Figure 6, ApoAI seems less susceptible when alone to non-enzymatic glycation. This is probably due to the presence of accessory proteins that should protect (at least to a certain extent) the HDL particle. This should be acknowledged in the text, because it could emphasize the detrimental effect of A-beta on HDL also in vivo.
8) The profiles in Figure 7, although interesting, seem all similar with or without A-beta and the redshift in the WMF is difficult to picture. So, I suggest to produce an emission spectrum each concentration of urea tested and for each mixture (with or without A-beta at different concentration). Then, the value of WMF could be summarized in a table or given in the text. In this context, the results from line 296 to 303 seem not appropriately presented.
In addition to this, it would be also nice to have the ratio intensities between solvent exposed and buried tryptophan fluorescent signal peak shapes, to better picture the transitions between folded and unfolded protein.
Finally, it would be interesting to titrate ApoAI with different concentrations of A-beta (treated as a denaturing agent like Urea), in the absence of urea. In this way (keeping in mind to correct for the autofluorescence of Abeta) the effect of A-beta on the stability of rHDL, if any, could be “bulletproof”.
9) In light of the flaws reported above, the results discussed from line 371 to 380 should be redone.
Minor revisions
General note: it is recommended that a Native English or a language service edits the manuscript. Few passages are difficult to follow due to language.
1) On page 1, line 39, please write: “More recently, coronary heart disease has been acknowledged as a potent risk factor for dementia.”
2) on page 2, line 86 please correct “The Hig-tagged..” with “ The His-tagged…”.
3) In the material and methods section, please provide the catalog number for all the used reagent in order to encourage replication of results.
4) On page 4 line 146, please specify DMPC what stands for.
5) line 248: please write “severely impaired by…”
6) on line 288, instead of “fructation” a generic “non-enzymatic glycation” would be more appropriate.
7) from line 310 to 312: the sentence is not well written and difficult to follow. Please rewrite it. The same is true for line 318 to 320.
8) the title on line 325 is not correct: please write “Toxicity on embryo and tissue regeneration of Abeta”.
9) from line 359 to 361: the sentence is not well written and difficult to follow. It should be rephrased.
Author Response

(The authors gave the same response as above.)

Round 2
Reviewer 1 Report
The authors have now amended the manuscript to my satisfaction
Reviewer 2 Report
I feel that the author replied to all my concerns.